# Purification of Camellia Oil by Inorganic Ceramic Membrane

**DOI:** 10.3390/foods11223644

**Published:** 2022-11-15

**Authors:** Danyu Cao, Lili Gai, Debao Niu, Yarong Li, Jianbin Li, Run Tian, Kai Li

**Affiliations:** 1College of Light Industry and Food Engineering, Guangxi University, Nanning 530004, China; 2Provincial and Ministerial Collaborative Innovation Center for Sugar Industry, Nanning 530004, China; 3Engineering Research Center for Sugar Industry and Comprehensive Utilization, Ministry of Education, Nanning 530004, China

**Keywords:** camellia oil, ceramic membrane, physicochemical properties, membrane flux

## Abstract

Camellia oil is an edible health oil with high medicinal value. While phospholipids, peroxides, and free fatty acids are present in unrefined camellia virgin oil (CVO), which has a negative impact on the quality characteristics and storage stability. This paper is to investigate the testing effects of transmembrane pressure and temperature on the membrane flux and degumming (the removal of colloidal substances from crude oil and which is mainly phospholipids) to determine the optimum process parameters for the purification of CVO. On this basis, the effects of purification treatments applied by using a membrane system with membranes of different pore sizes (200, 140, 20, 15, and 10 nm) on CVO were tested. The results indicate that the purification treatments of ceramic membrane on CVO reduced the contents of phospholipids (87.0% reduction), peroxides (29.2% reduction), and free fatty acids (16.2% reduction) at a transmembrane pressure of 0.4 MPa and temperature of 60 °C. At the same time, these treatments did not significantly alter the fatty acid composition. Thus, ceramic membranes have the potential for the purification of camellia oil, which could be an effective way to achieve the purification of camellia oil.

## 1. Introduction

Camellia oil is a high-quality edible oil extracted from the seeds of *Camellia sinensis* with fatty acid composition and physicochemical properties, which are very similar to olive oil [1,2]. In addition, camellia oil contains a variety of functional components such as phytosterols, squalene, vitamin E, and flavonoids [3]. These nutrients with multiple bioactivities can be used to lower blood pressure and cholesterol, protect the liver, fight against cancer as well as reduce gastrointestinal pain [4,5,6]. However, the content of these bioactive substances depends to a large extent on the extraction and processing technology, and the value of camellia virgin oil (CVO) is not fully reflected due to the limitations of the processing technology of camellia seeds [7].

Chemical refining of vegetable oils generally includes degumming, deacidification, decolorization, deodorization, and dewaxing [8], out of which degumming is an essential step [9]. At present, chemical refining by pressing production is still the main method for producing camellia oil in industry, which cannot avoid the disadvantages of high solvent consumption and organic solvent residues [10,11,12]. A number of green, efficient, and novel degumming processes, such as enzymatic purification [13], electrostatic field treatment, and physical sorbent purification, have been developed to overcome the drawbacks of conventional degumming, but they are expensive and/or complicated. In the enzymatic degumming developed by Lurgi, phospholipase A1 specifically catalyzes the release of phospholipids. It has the advantages of mild reaction conditions and a high reaction rate, but the high cost of the enzyme is not conducive to practical application [14,15]. Zhang et al. produced high-quality aromatic rapeseed oil by electrolytic degumming, and the final optimized degumming process (sodium chloride solution concentration of 3%, *w*/*w*) had a phospholipids removal rate of 90.7%; however, this approach was similar to aqueous degumming, improving only 5.10% compared to aqueous degumming, and still has the drawbacks of conventional degumming [16]. The physical adsorption method is simple and easy to operate, but trace components (e.g., tocopherols, phytosterols, and β-carotene) are lost together, thus reducing the functionality of the oil [17,18].

Furthermore, the increasing consumer demand for the high nutritional value of camellia oil has promoted the development of green purification technologies. It has been reported that membrane treatment of vegetable oils is also an emerging purification method [19]; however, there are few studies that have been performed on the membrane flux as well as the variations in the physicochemical properties of CVO during purification of the membrane, which has been developed for more than four decades since 1977 when Sen Gupta, A.K. published his patent: Refining Crude Oil by Filtration. Various membrane materials for purifying solvent oils/crude oils have been explored experimentally by different research teams in which they have achieved great success in the purification of the membrane [20]. So, purification of the membrane is performed by molecular size as well as molecular weight, and membrane materials are divided into organic and inorganic membranes [21]. In recent years, a great amount of work has been accomplished by several researchers to investigate mixed solvent oils because of the high viscosity of crude oil and low membrane flux. An asymmetric membrane of polyvinylidenfluoride can be used for effective solvent oil purification with measured retention values of phospholipids above 95% for the range of conditions studied (2 bar < ΔP < 6 bar; 30 °C < T < 50 °C), while the direct purification of the crude oil results in unsatisfactory fluxes [22]. Ultrafiltration membranes were used by Koris and Vatai [23] to achieve phospholipids retention of 70–77% (with a retention molecular weight of 15 kD), oil filtrations were carried out at temperature 40~60 °C, pressure 2~5 bar, and velocity of flow 0.3~0.4 L·m^−3^; however, vegetable oil permeates through the permeate at a very slow rate due to its high viscosity. In addition, the non-porous polymer composite hydrophobic membranes displayed very high phospholipids content reduction (over 99%) in undiluted oil, Experiments with undiluted oils were conducted at an operating pressure of 3 and 4 MPa in the membranes. Operating pressure for experiments with hexane-diluted oils was maintained at 2 MPa. All the experimental runs were conducted at room temperature (24~28 °C), while their oil flux was only 0.03 L·m^−2^·h^−1^ [24].

The objective of this study was to investigate the feasibility of the technology of ceramic membranes for the purification of CVO without solvent. To determine the optimal operating conditions, the process parameters of ceramic membranes with different pore sizes were evaluated. The quality analysis of camellia oil was carried out to research the removal of common contaminants (such as phospholipids, peroxides, and free fatty acids), as well as the changes in the fatty acid composition, were also investigated.

## 2. Materials and Methods

### 2.1. Materials

The ceramic membranes were purchased from Sterlitech Company (Kent, WA, USA). The specifications of the membranes are listed in Table 1. The micro-ceramic membrane system was provided by Shanghai Xiangming Biotechnology Co., Ltd. (Shanghai, China). CVO was supplied by Guangxi Zhongzhou Ecological Agriculture Investment Co., Ltd. (Hechi, Guangxi, China).

### 2.2. Purification Process

Figure 1 shows a schematic of the experimental setup. The ceramic membrane unit consists of a feed (5 L) tank connected to a variable feed pump and two manometers connected to the ends of the single-stage filtration ceramic membrane for the purification of CVO (2 L). CVO was driven by a raw material pump into the membrane filtration system and passed radially through the membrane. The retentate was returned to the feed tank, while the permeate was collected in a container located close to the device. The experiment was conducted by controlling a single variable; the TMP was initially selected from 0.1 MPa to 0.5 MPa, the T was initially selected from 20 °C to 70 °C based on extensive pre-experiments, and the optimum process parameters of both were determined by the membrane flux as well as the degumming rate (removal rate of phospholipids). The obtained purified camellia oil was collected while the retention solution was recycled back to the raw material liquid tank. Throughout the filtration process, the volume of CVO in the membrane system was kept constant at 2 L by the continuous addition of CVO to the feed tank. The membranes were timely cleaned after each experiment (with hot deionized water, followed by alternate backwashing with 2% NaOH and 0.5% HNO_3_, and finally with deionized water until the pH of the permeate was 7), and the water flux was measured to ensure membrane cleanliness.

The differential pressure across the membrane filtration process can be written as follows:(1)$$\mathrm{TMP}=\frac{P1+P2}{2}\;-\;P3$$
where TMP is the transmembrane pressure (MPa); P_1_ is the inlet pressure of the membrane module (MPa); P_2_ is the outlet pressure of the membrane module (MPa); P_3_ is the pressure on the permeate side (MPa).

The membrane permeation flux is defined as the volume of permeate flowing through a unit membrane filtration area per unit of time, which was calculated as follows:(2)$$J=\frac{V}{A_{0}\;\times \;\Delta t}$$
where J is the membrane permeation flux (L·m^−2^·h^−1^), V is the volume of the permeate (L), A_0_ is the membrane filtration area (m^2^), and Δt is the working time (h).

### 2.3. Determination of Phospholipids (PL) Content

The PL content of camellia oil is normally expressed in terms of phosphorus content. Phosphorus content was determined using molybdenum blues colorimetry following the AOCS official method Ca 12-55 (AOCS official method Ca 12-55, 2009) [25]. This method determines phosphorus by ashing the oil samples in the presence of zinc oxide, followed by the spectrophotometric measurement of phosphorus as a blue phosphomolybdic acid complex.

### 2.4. Fatty Acid Composition

The fatty acids were converted into the corresponding fatty acid methyl esters following the method of Moigradean with slight modifications [26]. Briefly, 100 μL of every prepared sample was diluted with 1 mL of hexane, filtered, and analyzed. The analysis was performed on a gas chromatograph GC 2010 Plus. In the GC system, a capillary column CD-2560 (SGE, Austin, TX, USA) with a length of 100 m and an inner diameter of 0.25 mm and 0.20 μm was used. The column temperature was started at 140 °C and held for 4 min, then heated from 140 °C to 230 °C at a rate of 2.5 °C·min^−1^, then held at 230 °C for 20 min, after which it continued to be heated from 230 °C to 250 °C at 5 °C·min^−1^ and held at 250 °C for 10 min (total program time 74.28 min). The injection temperature was maintained at 250 °C and the carrier gas was helium, the column flow was 0.5 mL·min^−1^, and 1 μL of each sample was injected with a separation ratio of 1:50. Spectra were recorded in full scan mode from 45 m·z^−1^ to 400 m·z^−1^ at 0.2 s·scan^−1^. The peaks generated by GC-MS were identified based on mass spectral characteristics and confirmed by mixing with a reference mixture of 37 fatty acid methyl esters for comparison and confirmation. The relative content of fatty acid in camellia oil samples was expressed as a percentage and determined by calculating the area of the corresponding fatty acid peaks and normalizing the sum of the areas of all fatty acid peaks as appropriate.

### 2.5. Physicochemical Characteristic Analysis

The camellia oil samples obtained were characterized according to the American Oil Chemists’ Society. Experimental measurements of viscosity were performed using a rheometer with a 40 mm 2° tapered plate fixture and a parallel plate fixture [27]. The moisture and volatile material content were evaluated by the vacuum oven method (AOCS, Ca 2d-25, 2017) [28], The acid value of CVO was quantified by the method (AOCS, Ca 3d-63, 2017) [29], and the peroxide value (AOCS, Cd 8-53, 1998) was carried out to evaluate the oxidation [30]. The refractive index was determined by Abbe’s refractometer [31]; the saponification values were determined using direct titration (AOCS, Ca 3-25, 2017) [32]; UV absorbance was calculated from the coefficient of absorption at 232 nm [33].

### 2.6. Data and Statistical Analysis

Analyses were performed in triplicate, and the results were subjected to an analysis of variance for a completely random design and expressed as means ± standard deviation. Statistically significant differences were determined by using SPSS version 27.0, and *p* < 0.05 was considered to be statistically significant.

## 3. Results and discussion

### 3.1. Influence of Operating Conditions for Purification on Membrane Flux and Degumming Rate

The experiments were conducted to verify the effect of the membranes with different pore sizes, and the operating conditions (TMP and T) on the permeate flux are reported below. The membrane fluxes obtained at different TMPs were first analyzed (T 60 °C), as shown in Figure 2a. TMP is the sole driving force of the purification process of ceramic membranes and has a decisive influence on the membrane flux. Although the influence of TMP varies with the physical and chemical properties of the feed fluid, it is always the primary factor in determining the membrane flux during the purification process of ceramic membranes. It can be seen from Figure 2a that the membrane flux of CVO by ceramic membrane was significantly changed (*p* value) at different operating pressures. The purification process of CVO by ceramic membrane can be roughly divided into two stages. Stage I: The membrane flux increased linearly with TMP from 0.10 to 0.40 MPa as expected. Stage II: The membrane fluxes of different pore sizes exhibit a slow increase or remain constant at TMP above 0.40 MPa. This fact is probably linked to the fouling phenomenon, which occurs at higher pressure [34]. The high pressure may cause some of the smaller molecular weight PL micelles to be rapidly squeezed through the membrane, resulting in serious blockage of the pores of the ceramic membrane and causing the flux to stabilize or decline.

Figure 2b shows the variation of membrane flux with T (TMP 0.4 MPa). As compared to Figure 2a, the effect of T is less relative to the effect of TMP on membrane flux. The membrane flux increased with T in the temperature range of 20 °C~60 °C. Operating at high temperatures is beneficial for flux efficiency as it enhances mass transfer and thus permeability [35]; on the other hand, this was also expected due to the decrease in viscosity or the increase in PL diffusion on the pores of the membrane. In this temperature range, the membrane flux of the ceramic membrane (10 nm) is increased by a factor of 4 (1~5 L·m^−2^·h^−1^). However, a further temperature increase (60 to 70 °C) led to a decrease or slow growth in flux, confirming that the T of 60 °C was suitable for the purification of camellia oil; above the T of 60 °C, the flux decreased. A decline in membrane flux is predicted because of the fouling on the membrane surface because of gelatinization [36].

Based on the results of extensive pre-experiments on degumming, the ceramic membrane with a pore size of 10 nm was selected for further studies on the effect of TMP and T on the degumming of camellia oil. Figure 2c shows the effect of TMP on the degumming rate at 60 °C. The increase in applied pressure had no significant influence on PL rejection for the membrane tested; these results were similar to those reported by García et al. [37]. When TMP was 0.4 MPa, the degumming rate of 86.98% was the maximum. However, there was a significant decrease in the degumming rate when the TMP exceeded 0.40 MPa. This is probably due to the high operating pressure causing some small molecular weight PL micelles to be rapidly squeezed across the membrane resulting in a decreased degumming rate. The effect of T on the degumming rate at a TMP of 0.4 MPa is shown in Figure 2d. The degumming rate increased for the membrane with increasing T and the degumming rate at 60 °C was about 9.07% higher than that at 20 °C, which tended to level off when T rose above 60 °C. However, there was a significant decrease in the degumming rate when the TMP exceeded 0.40 MPa.

The best conditions for carrying out a purification process with these ceramic membranes would be those that lead to a low PL (high rejections of the components). Furthermore, a high membrane flux is desirable. As a result of the experiments, the best conditions chosen for a purification process with the ceramic membranes were the following: TMP, 0.4 MPa; T, 60 °C.

### 3.2. Phospholipids (PL) Composition Analysis

As we all know, triglycerides account for more than 95% of raw vegetable oils, and the rest of the components include PL, free fatty acids, carbohydrates, proteins, degradation products, etc. [38]. One of the most important influences on the quality of vegetable oils is the PL; therefore, degumming during the purification process is particularly important. Removing PL from vegetable oils by passage through the ceramic membrane has previously been reported, which was further confirmed in this study. The degumming of ceramic membranes is an emerging separation process for the degumming of CVO with the help of selective permeation of ceramic membranes for various components under the action of a driving force (TMP). Theoretically, it is difficult to separate triglycerides and PL with ceramic membranes due to the molecular weight difference between them being very small (molecular weight of about 900 Da) [39]. De Moura et al. performed experiments on the removal of PL from crude soybean oil with a maximum removal rate of 77% at a permeate flow rate of 5 L·m^−2^·h^−1^. They also proposed that a polarized layer is formed, and subsequently, PL are deposited on the membrane surface as filtration proceeds. This gel layer formed on the membrane surface is often referred to as a “dynamic membrane” [40]. PL are surfactants with hydrophobic and hydrophilic groups that tend to aggregate [41]. When CVO flowed at high speed on the membrane surface, PL molecules aggregated to form macromolecules with much higher molecular weight than triglycerides, which were trapped by the membrane pores and deposited on the membrane surface.

The results of the PL contents of six kinds of camellia oil samples (CG: CVO, A: 200 nm, B: 140 nm, C: 20 nm, D: 15 nm, E: 10 nm) can be shown in Figure 3, the degumming effect of 200 nm and 140 nm on CVO was not satisfactory with degumming rates of only 30.15 and 34.33%, and the degumming rate was still low (40.54%) when the pore size of the ceramic membrane was reduced (20 nm). However, the degumming rate was significantly increased (80.50%) when the pore size of the ceramic membrane continued to shrink (15 nm). The smallest pore size of the ceramic membrane (10 nm) achieved a degumming rate of 86.07%, which means that the phosphorus content was decreased from 61.80 to 8.61 mg·kg^−1^ with a membrane flux of 4.22 L·m^−2^·h^−1^. Similarly, a significant increase in degumming rate using tubular ceramic membranes (Al_2_O_3_/TiO_2_—pore size 0.01 μm) during the treatment of crude mustard oil has been shown by Alicieo et al., while the phosphorus content was decreased from 463.32 to 3.97 mg·kg^−1^ with a 99.14% degumming rate [42]. The lower degumming rate in this experiment compared to its lower rate may be due to the difference in degummed raw oil and the phosphorus content contained in CVO (61.80 mg·kg^−1^) was much lower than that in crude mustard oil (463.32 mg/kg).

### 3.3. Fatty Acid Composition Analysis

Fatty acid composition is a key parameter in vegetable oils since it represents an essential indicator of their nutritional value on the one hand and their stability on the other hand. The camellia oil fatty acid composition is a good source of oleic acid, which is a monounsaturated fatty acid of the omega-9 family [43,44]. Table 2 summarizes the results of the relative percentage of fatty acids of the purified camellia oil samples obtained during the membrane processing of CVO. The contents of oleic acid and linoleic acid in CVO reached 82.20 and 7.67%, respectively, especially linolenic acid which accounted for 0.24% and is an essential fatty acid for the human body. As can be seen from Table 2, the unsaturated fat contents of different camellia oil samples were above 90%, out of which the unsaturated fatty acid content of CVO was about 90.76%, and the unsaturated fatty acid contents of A, B, C, D, and E was approximately 90.73, 90.59, 90.72, 90.60, and 90.14%. At the same time, analysis of all other fatty acids showed a slight decrease in the total unsaturated fatty acid content (decreased by 0.03~0.68%). Palmitic, oleic, and linoleic fatty acids accounted for more than 90% of the glycerol ester fatty acid fraction in CVO. For all fatty acids, the purifying process did not induce any significant modification of the fatty acid’s distribution (relative percentage of fatty acids). The difference in molecular size between the major molecular species of camellia oil triglycerides is only~30 Da; the differences in molecular size and polarity between these triglycerides were not sufficient to induce membrane selectivity, which shows that the strong fatty acid balance of camellia oil is not affected in this process.

### 3.4. Physicochemical Characteristic of Camellia Oil Analysis

Viscosity is an important physical property parameter of oils, which is significant to the design of transport, heat transfer, mass transfer, and separation equipment during oil processing. The viscosity of CVO is closely related to the size and structure of the molecules. Secondly, the viscosity is also closely related to the temperature.

The effect of purification by ceramic membrane on the viscosity of CVO with temperature is determined by the shear rate. As shown in Figure 4a, the effect of shear rate on the viscosity of CVO was tested at five different temperatures. The viscosity of CVO remained constant with increasing shear rate at each temperature, indicating that the viscosity of CVO was not affected by the shear rate. It can be observed that the viscosity fluctuated more when the shear rate was from 0 s^−1^ to 10 s^−1^, while the viscosity was stable when the shear rate was at 10 s^−1^. As a result, the shear rate of 10 s^−1^ can be chosen as the experimental condition.

The change in viscosity of oil with temperature is called viscosity–temperature performance, and the performance of high viscosity has a smaller range with temperature. As shown in Figure 4b, the viscosity of camellia oil tended to increase slightly after purification by a ceramic membrane at a shear rate of 10 s^−1^ and temperatures of 0 °C~120 °C. Sipos suggested an inverse relationship between viscosity and unsaturation of fatty acids [45]. At the same time, there is a positive relationship between viscosity and PL content [46]. It was experimentally verified that the percentage of unsaturated fatty acids and PL content were reduced after the purification of CVO by a ceramic membrane; therefore, the unsaturation of fatty acids has a greater effect on the viscosity of camellia oil than the effect of PL on viscosity.

The physicochemical properties of camellia oil obtained by purification of the ceramic membrane are the most common and important quality evaluation indicators in the edible oil industry. One of the main advantages of membrane treatment is that oxidative changes can be avoided because of the mild operating conditions. Figure 5a shows the amount of oxidation products in camellia oil with peroxide value (POV) reflecting the formation of hydroperoxides caused by primary oxidation. In addition, some peroxides present in the raw material were significantly rejected during the membrane purification process, as indicated by the lower POV in the purified camellia oil, with a maximum reduction of 29.18%. Moreover, a lower acid value was also observed, with a maximum reduction of 16.47% (Figure 5b), which was not a significant reduction. This is expected because the permeability of free fatty acids is similar to that of triglycerides during the purification process of ceramic membrane and the free fatty acids content in the crude oil itself is very low, which leads to little effect on the retention of free fatty acids. The decrease in acid value observed in the experiment is possibly due to the entrapment of PL [47].

The UV absorbance was lower for crude and unrefined pressed oils and higher for refined vegetable oils. From Figure 5c, it can be seen that the overall UV absorbance of camellia oil shows an increasing trend after purification by the ceramic membrane, indicating that the quality of purified camellia oil was improved. As shown in Figure 5d–f, the refractive index of vegetable oil is closely related to the composition and structure of the oil and can be used to identify the type and purity of the oil. The results of the refractive index of camellia oil samples were also within the established range (1.466–1.470) without significant changes, and the saponification and moisture values of camellia oil likewise did not change visibly, and these values were similar to those of CVO. From the above analysis of the physicochemical properties of camellia oil, it is clear that the purification by ceramic membrane does not destroy the quality of camellia oil but rather has a good improvement on the quality.

## 4. Conclusions

This experiment investigated the feasibility of the purification of CVO using inorganic ceramic membranes. The results indicate that the optimal parameters for maximum purification effect in CVO were found to be 40 °C as T and 0.4 MPa as TMP. The phosphorus content of the oil was reduced from 61.80 to <9 mg·kg^−1^ under the optimal degumming process. Purification reduced POV and AV but did not impact the composition of fatty acids. Further, the refractive index, saponification value, and moisture did not change significantly due to purification. A series of quality measurements show that the purified camellia oil preserves the main physicochemical and nutritional properties. Furthermore, the purification of camellia oil by ceramic membranes needs further research, and one of the biggest problems is that the current membrane flux is not ideal for industrialization.

## Figures and Tables

**Figure 1 foods-11-03644-f001:**
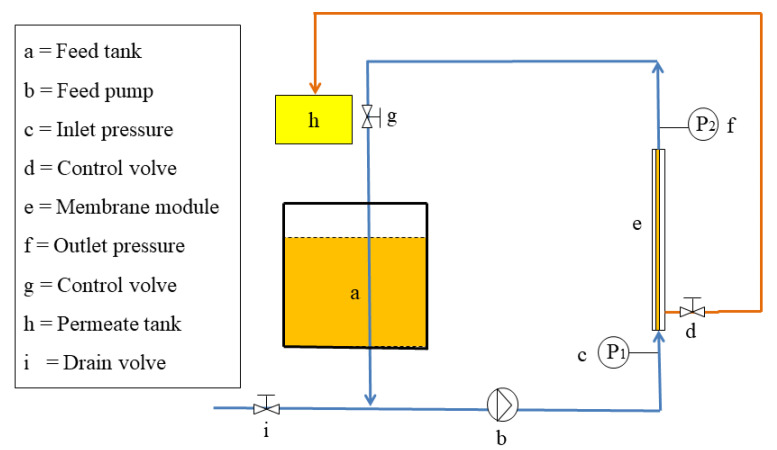
Experimental device of the ceramic membrane system.

**Figure 2 foods-11-03644-f002:**
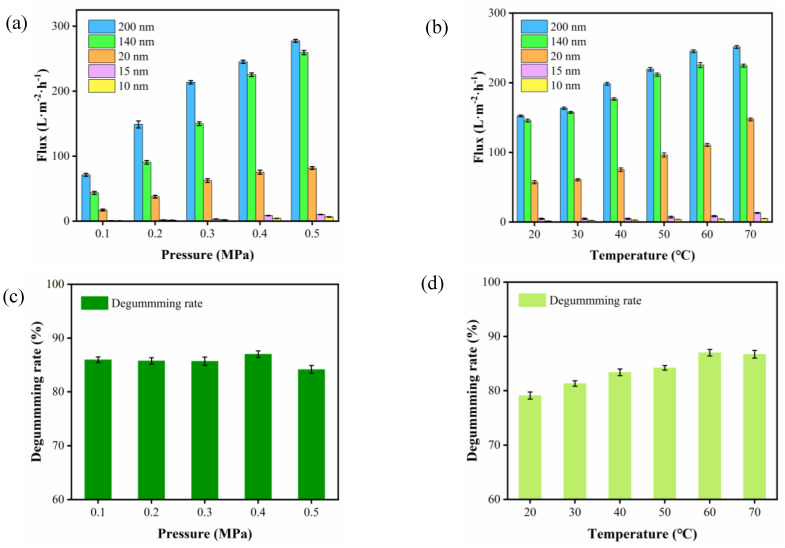
(**a**) Effect of different TMP on membrane fluxes at 60 °C. (**b**) Effect of different temperatures on membrane fluxes at 0.4 MPa. (**c**) Effect of different TMP on the degumming rate at 60 °C. (**d**) Effect of different temperatures on the degumming rate at 0.4 MPa.

**Figure 3 foods-11-03644-f003:**
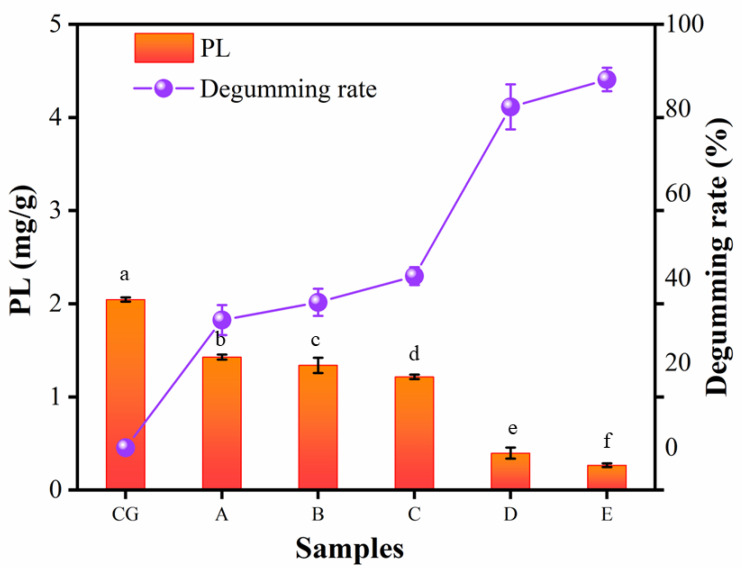
The PL retention distribution of single-stage permeate filtration using inorganic ceramic membranes was controlled at 0.4 MPa and 60 °C: (CG) CVO; (A) 200 nm; (B) 140 nm; (C) 20 nm; (D)15 nm; (E) 10 nm. A different lowercase means a significant difference (*p* < 0.05) among samples. Means and standard deviations (SD) were obtained from three replicates; the degumming rate was calculated with the averages.

**Figure 4 foods-11-03644-f004:**
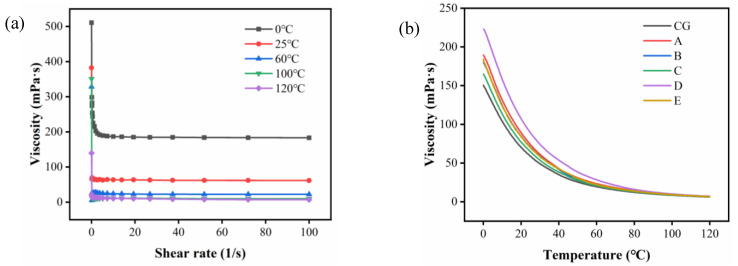
(**a**) Viscosity versus shear rate curve of different temperatures; (**b**) effect of temperature on the viscosity of different camellia oil samples: (CG) CVO; (A) 200 nm; (B) 140 nm; (C) 20 nm; (D)15 nm; (E) 10 nm.

**Figure 5 foods-11-03644-f005:**
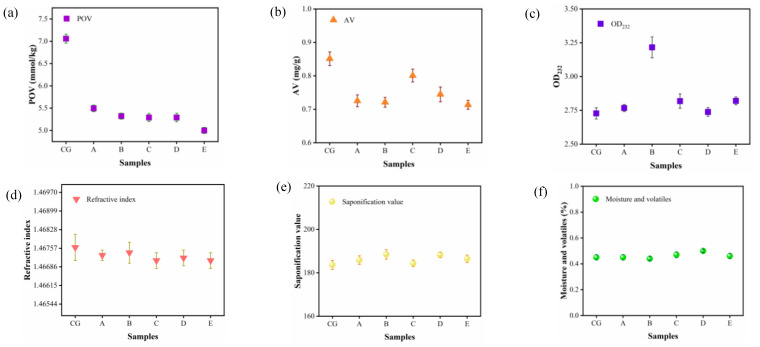
Effect of ceramic membrane degumming on peroxide value (**a**), acid value (**b**), UV absorbance (**c**), refractive index (**d**), saponification value (**e**), and moisture content of camellia oil (**f**). (CG) CVO; (A) 200 nm; (B) 140 nm; (C) 20 nm; (D)15 nm; (E) 10 nm.

**Table 1 foods-11-03644-t001:** Material characteristics and module details of the membrane system.

Nomenclature	Description
Manufacturer	TAMI Separation Technologies (Suzhou Industrial Park) Co.
Membrane type	Tubular
Activated microfiltration separation layer	ZrO_2_
Activated ultrafiltration separation layer	TiO_2_
Membrane support material	Al_2_O_3_/TiO_2_/ZrO_2_
Pore size (nm)	200, 140, 50, 15, 10
Pure water permeability (25 °C, 0.1 MPa)	468.44, 434.28, 218.28, 64.89, 32.70
Length (cm)	25
Outer diameter (mm)	6
Channel diameter (mm)	5.5
Number of channels	1
Surface area (m^2^)	0.0043175
Operating pH	0~14
Maximum operating temperature (°C)	350

**Table 2 foods-11-03644-t002:** Fatty acid composition analysis (%).

Fatty Acids	Samples (Permeate)					
	CG	200 nm	40 nm	20 nm	15 nm	10 nm
ethyl oleate (C18:1)	82.2 ± 0.033 ^a^	82.2 ± 0.042 ^a^	82.0 ± 0.065 ^b^	82.2 ± 0.026 ^a^	82.0 ± 0.0054 ^b^	81.6 ± 0.02 ^c^
Methyl linoleate (C18:2)	7.67 ± 0.028 ^ab^	7.65 ± 0.03 ^ab^	7.69 ± 0.05 ^ab^	7.63 ± 0.018 ^ab^	7.70 ± 0.035 ^a^	7.62 ± 0.042 ^b^
Methyl palmitate (C16:0)	7.46 ± 0.020 ^c^	7.46 ± 0.0018 ^c^	7.59 ± 0.0021 ^b^	7.46 ± 0.024 ^c^	7.62 ± 0.053 ^b^	7.88 ± 0.06 ^a^
Methyl stearate (C18:0)	1.65 ± 0.009 ^b^	1.65 ± 0.023 ^b^	1.69 ± 0.024 ^ab^	1.67 ± 0.10 ^ab^	1.65 ± 0.057 ^b^	1.77 ± 0.026 ^a^
Methyl 11-eicosenoate (C20:1)	0.416 ± 0.02 ^a^	0.411 ± 0.07 ^a^	0.415 ± 0.023 ^a^	0.412 ± 0.018 ^a^	0.410 ± 0.014 ^a^	0.410 ± 0.020 ^a^
Methyl linolenate (C18:3)	0.243 ± 0.0024 ^ab^	0.240 ± 0.0009 ^c^	0.245 ± 0.001 ^a^	0.245 ± 0.0014 ^a^	0.240 ± 0.0018 ^c^	0.242 ± 0.001 ^bc^
Methyl palmitoleate (C16:1)	0.068 ± 0.002 ^e^	0.069 ± 0.0028 ^de^	0.084 ± 0.0024 ^c^	0.071 ± 0.0018 ^d^	0.087 ± 0.0014 ^b^	0.110 ± 0.0012 ^a^
Methyl nervonate (C24:1)	0.056 ± 0.0018 ^a^	0.056 ± 0.0023 ^a^	0.057 ± 0.0022 ^a^	0.055 ± 0.0018 ^a^	0.055 ± 0.0011 ^a^	0.055 ± 0.0014 ^a^
Methyl heptadecanoate (C17:0)	0.054 ± 0.0014 ^a^	0.053 ± 0.0024 ^a^	0.055 ± 0.0015 ^a^	0.054 ± 0.0020 ^a^	0.053 ± 0.0023 ^a^	0.055 ± 0.0024 ^a^
Methyl heptadecenoate (C17:1)	0.052 ± 0.0015 ^a^	0.054 ± 0.0013 ^a^	0.054 ± 0.001 ^a^	0.053 ± 0.002 ^a^	0.053 ± 0.0017 ^a^	0.054 ± 0.0013 ^a^
Methyl arachidate (C20:0)	0.029 ± 0.0014 ^a^	0.029 ± 0.0017 ^a^	0.030 ± 0.002 ^a^	0.029 ± 0.0024 ^a^	0.028 ± 0.0012 ^a^	0.031 ± 0.0010 ^a^
Methyl myristate (C14:0)	0.028 ± 0.002 ^bc^	0.026 ± 0.0015 ^c^	0.030 ± 0.0013 ^b^	0.029 ± 0.0017 ^b^	0.028 ± 0.0012 ^bc^	0.038 ± 0.0018 ^a^
Methyl erucate (C22:1)	0.028 ± 0.0023 ^a^	0.028 ± 0.001 ^a^	0.028 ± 0.001 ^a^	0.028 ± 0.0027 ^a^	0.029 ± 0.002 ^a^	0.026 ± 0.0034 ^a^
Methyl lignocerate (C24:0)	0.025 ± 0.005 ^a^	0.026 ± 0.0029 ^a^	0.025 ± 0.003 ^a^	0.026 ± 0.0017 ^a^	0.024 ± 0.003 ^a^	0.025 ± 0.004 ^a^
Methyl Docosahexaenoate (C22:6)	0.022 ± 0.001 ^bc^	0.020 ± 0.001 ^c^	0.021 ± 0.0005 ^c^	0.024 ± 0.0023 ^ab^	0.024 ± 0.0017 ^ab^	0.025 ± 0.0021 ^a^
Methyl behenate (C22:0)	0.012 ± 0.0011 ^a^	0.011 ± 0.0013 ^ab^	0.011 ± 0.0017 ^ab^	0.010 ± 0.002 ^b^	0.011 ± 0.001 ^ab^	0.012 ± 0.001 ^ab^
Methyl pentadecanoate (C15:0)	0.005 ± 0.0002 ^c^	0.006 ± 0.0002 ^b^	0.007 ± 0.0002 ^a^	0.006 ± 0.0002 ^b^	0.006 ± 0.0001 ^b^	0.007 ± 0.0003 ^a^

Note: Results are mean ± SD of triplicate determinations. Different letters in the same line indicate significant differences (*p* < 0.05).

## Data Availability

Data are contained within the article.

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
