# Peer review of "Purification of Camellia Oil by Inorganic Ceramic Membrane"

_foods, 2022, doi:10.3390/foods11223644_

Round 1
Reviewer 1 Report
The subject investigated is of interest to the readers. The manuscript is very well written, however, minor amendments are required to improve the overall quality.
A complete report on the manuscript is attached.

Reviewer 2 Report
Manuscript n. “foods-1956948-peer-review”
Comments:
The manuscript is a report of the results obtained in the purification of camellia virgin oil by inorganic ceramic membranes.
Membrane performances in the degumming process are reported and the quality of the purified oil is documented, included viscosity measurements. As a whole, the manuscript is interesting.
The “materials and methods” section is clear and well detailed with reference to all the analytical techniques. However, there is a total lack of information about the characteristics of the membranes/modules used. None of the experiments can be reproduced by the reader.
The detailed information must be provided on the membranes and modules used.
Detailed comments
Abstract: please revise English.
Materials and methods.: 1) membrane characteristics should be provided: material, number of layers, porosity and thickness of each layer; 2) module characteristics: shape (flat disc or tubular), geometry( disc diameter, or inner and outer diameter of the tube and the tube length), membrane area. 3) membrane performances: nominal rejection or MWCO and water flux. And so on.
Figure 2. Please check the y-axis title of (a) and (b): what does “aperture” represent?
The clear definition of “degumming” should be provided.
What is the definition of “degumming rate” ?
Table 1: please specify in the title if the samples are of retentate or of permeate. Symbols used of fatty acids is certainly usual, however the addition of the name would be very appreciated.
Figure 4. Please, clarify the caption. “..viscosity of different ceramic membranes…” does not make sense.
Round 2
Reviewer 2 Report
accept